# Estimation of PM_10_ Levels and Sources in Air Quality Networks by Digital Analysis of Smartphone Camera Images Taken from Samples Deposited on Filters

**DOI:** 10.3390/s19214791

**Published:** 2019-11-04

**Authors:** Selena Carretero-Peña, Lorenzo Calvo Blázquez, Eduardo Pinilla-Gil

**Affiliations:** Department of Analytical Chemistry and IACYS, University of Extremadura, Av. de Elvas, s/n, 06006 Badajoz, Spain; selenacarretero@unex.es (S.C.-P.); lorcalvo@unex.es (L.C.B.)

**Keywords:** atmospheric particulate matter, air quality networks, smartphone camera, digital image analysis

## Abstract

This paper explores the performance of smartphone cameras as low-cost and easily accessible tools to provide information about the levels and origin of particulate matter (PM) in ambient air. We tested the concept by digital analysis of the images of daily PM_10_ (particles with diameters 10 µm and smaller) samples captured on glass fibre filters by high-volume aerosol samplers at urban and rural locations belonging to the air quality monitoring network of Extremadura (Spain) for one year. The images were taken by placing the filters inside a box designed to maintain controlled and reproducible light conditions. Digital image analysis was carried out by a mobile colour-sensing application using red, green, blue/hue, saturation, value/hue, saturation, luminance (RGB/HSV/HSL) parameters, that were processed through statistical procedures, directly or transformed to greyscale. The results of the study show that digital image analysis of the filters can roughly estimate the concentration of PM_10_ within an air quality network, based on a significant linear correlation between the concentration of PM_10_ measured by an official gravimetric method and the colour parameters of the filters’ images, with better results in the case of the saturation parameter (S_HSV_). The methodology based on digital analysis can discriminate urban and rural sampling locations affected by different local particle-emitting sources and is also able to identify the presence of remote sources such as Saharan dust outbreaks in both urban and rural locations. The proposed methodology can be considered as a useful complement to the aerosol sampling equipment of air quality network field units for a quick estimation of PM_10_ in the ambient air, through a simple, accessible and low-cost procedure, with further miniaturization potential.

## 1. Introduction

Particulate matter (PM) is well known as one of the most threatening atmospheric pollutants on human health condition, together with ozone, nitrogen dioxide and sulphur dioxide [1], PM_2.5_ (particles less than 2.5 µm diameter) and PM_10_ (less than 10 µm diameter) being of special concern. PM is emitted from a wide set of sources, mainly from industrial production and traffic, as well as by natural sources like soil resuspension by wind, forest fires or desert dust outbreaks [2]. Therefore, monitoring PM pollution and controlling their concentration in ambient air have become one of the main objectives of environmental protection agencies and public health protection organizations.

Currently, surveillance of PM levels to comply with health protection regulations (e.g., EU Directive 2008/50/EC on ambient air quality and cleaner air for Europe) is carried out through sophisticated instrumentation installed in field units within air quality monitoring networks. The measurements are performed either by advanced automatic equipment for continuous detection of particulate matter concentration based on beta-attenuation technique, light scattering or tapered element oscillating microbalance [3,4,5], or by the official method (EN 12341:2014) that consists in the PM_10_ or PM_2.5_ collection on glass or quartz fibre filters by automatic high-volume samplers, followed by gravimetric analysis. These methodologies provide reliable results; however, the required equipment is expensive and bulky, requiring conditioned environment (e.g., controlled temperature) and continuous maintenance by specialized technicians. Research efforts have been directed in recent years to the development and commercialization of reduced-price optical particle counters [6,7] that are more affordable to official institutions, research centres and even citizens. These technologies are in the early stages of development and validation, so they are presently not sufficiently qualified to replace reference instruments, according to a recent report of the World Meteorological Organization [8]. In this context, research is needed to find innovative procedures for PM measurements that include, among other features, minimizing analysis time, reducing maintenance costs and operation, simplifying the methodology so it can be applied by non-specialized personnel and building portable and miniaturized equipment for the measurement of PM levels in different locations.

Smartphone cameras have emerged in recent years as innovative components of analytical instrumentation. Specifically, some groups have explored image analysis capabilities of integrated smartphone cameras in a range of scientific studies focussed on determining analyte concentration in different kinds of samples, by monitoring colour changes due to chemical reactions. As representative examples, a digital camera has been used to determine the Ni content in steels [9] or to analyse Ca in water samples [10]. On the other hand, integrated smartphone cameras have been used to analyse the concentration of several elements [11], monitoring fluoride content in water [12], estimating the time deposition of bloodstains in a crime scene [13], determining the concentration of the Allura red dye in candies [14] or quantifying trinitrotoluene in soils [15]. These approaches make use of appropriate image analysis software, such as ImageJ, or mobile applications created to classify the colours based on colour spaces (red, green, blue (RGB), hue, saturation, value (HSV)) or colour patterns established by convention. The expanding analytical potential of smartphone spectrometers has been recently reviewed [16].

Regarding the use of smartphone cameras for estimating PM levels in ambient air, the literature contains only scarce works intended to use the digital analysis of sky pictures as a source of physico-chemical information about atmospheric aerosol. In this sense, a novel methodology has been developed [17], in which a special lens was attached to the front of the smartphone camera to convert it into a portable spectropolarimeter that could estimate particulate matter levels in the air through the aerosol optical thickness (AOT). It provided quasi-real-time information about the atmospheric aerosol with spatiotemporal coverage that complemented the information obtained by conventional equipment, using a network of active citizen scientists who obtained and transferred the data through a mobile application. With a similar strategy, Liu et al. [18] reported a method to estimate PM air pollution based on the analysis of outdoor images in locations in China and the USA. Six image features extracted from the images were used together with other relevant data, such as the position of the sun, date, time, geographic information and weather conditions, to predict PM_2.5_ indexes.

Another low-cost alternative for estimating the degree of air pollution by specific particles or fractions (e.g., black smoke (BS), black carbon (BC) or light absorbing carbon (LAC)) consists in the measurement of the interaction of radiation with filters, where the atmospheric particles have been deposited by means of aspiration using optical systems of very different characteristics based on dedicated instruments operated by reflectometry and/or transmissometry. The measurement of the transmitted radiant energy is the most generalized approach [19,20], and it has been used profusely in air quality monitoring networks to estimate the levels of black carbon [21]. To further simplify the instrumentation to obtain data on particulate matter deposited on filters, some researchers have explored the use of low-cost colorimeters and even smartphone cameras as tools to measure the interaction of the radiation emitted by a source and the particulate matter deposited. For example, Ramanathan et al. [22] proposed a system that integrates a miniaturized aerosol filter sampler with a cell phone for filter image collection, transmission and image analysis for determining BC in real time. Olson et al. [23] developed a system that quantifies colour coordinates of atmospheric particles deposited on filters to estimate the levels of elemental carbon (EC) and organic carbon (OC).

In this work, we aimed to explore the use of smartphone cameras as a tool for roughly estimating PM levels through the simplest approach of taking pictures of filters with a smartphone camera under a controlled light environment and performing a digital analysis of the image, as a possible tool for a preliminary estimation of particulate matter pollution levels and sources. To our knowledge, this is the first report about the use of a smartphone camera as a tool for the direct estimation of PM levels based on colorimetric measurements of the aerosol collected on filters.

## 2. Materials and Methods

### 2.1. Instruments and Software

All PM_10_ samples were obtained on 15 cm diameter glass fibre filters using automatic high-volume (30 m^3^/h) Digitel DAH-80 air samplers equipped with a PM_10_ head inlet (Digitel Elektronik AG, Hegnau, Switzerland) placed in field air quality monitoring units. A high-precision balance (10 µg Mettler Toledo, S.A., model AX205 Delta Range, Greifensee, Switzerland) was used to perform the standard gravimetric PM_10_ measurements by the reference method. Filter images were obtained using a smartphone BQ model Aquaris M5 (Worldreader S.L., Madrid, Spain), equipped with a Sony IMX214-13 megapixel (MP) camera, 5 lenses and 1/3.06″ sensor. The image processing was carried out using the mobile application Color Grab™, version 3.6.1 (Loomatix Ltd., Munchen, Germany, www.loomatix.com). The mathematical and graphic analyses of the results were carried out using the XLSTAT 2016-02 (Addinsoft Software, New York, NY, USA) and GNUPLOT 5.2 Patchlevel 0 (Slashdot Media, La Jolla, CA, USA, www.gnuplot.info) software.

### 2.2. Sample Collection

Daily atmospheric particulate matter samples (PM_10_ fraction) were collected from 1 January to 31 December, 2015 (total of 350 samples) at two monitoring stations belonging to the Air Quality Monitoring Network of Extremadura (REPICA) located in the city of Badajoz and in the Monfragüe National Park (Figure 1). Badajoz (150,543 inhabitants) is the most populous city in the region, characterized by a low industrial but intense commercial activity and considerable traffic with Portugal and other populated surrounding areas. The air quality monitoring station in Badajoz is placed in a suburban area within the campus of the University of Extremadura (LAT: 38°53′12″ N, LONG: 6°58′15″ W). The Monfragüe National Park is located in the province of Cáceres, in a natural environment of Mediterranean forest, meadow and rocky areas crossed by the Tajo and Tiétar rivers; it is characterized by containing a large number of animal species and a large tourist influx each year (around 300,000 visitors). The air quality monitoring unit of Monfragüe is placed in a remote area of the Park (Las Cansinas, LAT: 39°50′37″ N, LONG: 5°56′30″ W). The two air quality monitoring units were selected in order to maximize air pollution variability within the region, to test the applicability of the new digital image analysis methodology on PM_10_ filters captured under very different environments.

### 2.3. PM_10_ Gravimetric Reference Method

The PM_10_ mass concentration in air was measured according to the standard reference method EN 12341:2014 [24]. The filters were conditioned, before and after sampling, in a gravimetric cabin under controlled conditions of temperature (20 ± 1 °C) and relative humidity (45–50%), at least 48 h before weighing on a high-precision analytical balance. The difference between the two weighings obtained gives the PM_10_ particles’ net mass. This mass divided by the sampled air volumes provides the aerosol mass concentration in ambient air expressed as µg/m^3^.

### 2.4. Image Acquisition System and Procedure

Figure 2 illustrates the low-cost photographic system designed to obtain reproducible images of PM_10_ samples collected on the sampling filters as explained above. The system was composed of a conventional smartphone and an opaque white 25 × 25 × 25 cm wood box (Figure 2a). The box was designed in order to provide stable lighting conditions, which was achieved by using a stable LED strip that provided light of equal intensity and colour as natural light from the sun (LED 220 VAC SMD5050 silicone/epoxy, 60 LEDS Epister 10 W, 800 lumens, IP65, 6500k cold white) as shown in Figure 2b. A plastic circular 15 cm diameter sample holder was fixed in the lower central part inside the box (Figure 2b) to hold the PM_10_ filters, so that the centre of the sample corresponded to the viewing window located in the upper part of the box set to fit the smartphone camera properly (Figure 2c).

For the image acquisition, PM filters were placed in the sample holder (Figure 2b), the box was closed by a lock, the smartphone was placed on the viewing window and the LED lamp was turned on. The smartphone’s camera was programmed with optimal exposure conditions, namely, autofocus to 4 mm, automatic white balance, ISO 100 sensitivity, time of exposure 1/50 s and flash off. Photographs of each filter were recorded in triplicate and stored in the smartphone’s memory in .jpg format, 24-bit RGB system and 4160 × 3120 pixel resolution.

### 2.5. Image Processing and Analysis

The description of the colour in the three different colour models (RGB, HSV and hue, saturation, luminance (HSL)) was determined by applying the mobile application (app) Color Grab to the images taken from the filters. The images were loaded by the app from the smartphone’s memory, selecting the central region of the image (Figure 3a) and extracting the information from the RGB/HSV/HSL colour of the region (Figure 3b), as values of R (red), G (green), B (blue), Hue (H_HSV_, H_HSL_), saturation (S_HSV_, S_HSL_), value (V) and luminance (L). Colour parameter values were stored as averages of three replicates.

We also worked in the greyscale system, which allows users to obtain a single parameter of the colour intensity based on the combination of RGB colour space values, facilitating and simplifying colour analysis. Conversion of RGB to greyscale space can be done by three methods with different algorithms [25,26]—method of luminosity (Equation (1)), method of average (Equation (2)) and method of lightness (Equation (3)):
I = 0.21R + 0.72G + 0.07B,(1)
I = (R + G + B)/3,(2)
I = (max (R, G, B) + min (R, G, B))/2.(3)


The luminosity method (Lu) considers the degree of appreciation of each colour by the human eye, so RGB coefficients are different from each other and superior to the G parameter, since green is the most sensitive colour to the human eye. For this reason, it is the main methodology for the conversion of RGB system to greyscale. The average method (Avg) is the simplest and the lightness method (Li) considers the colours more and less intense. In this work, the values of colour intensity in greyscale using three methods were calculated, in order to compare the results obtained with each and to establish which of them provides the best results.

### 2.6. Saharan Dust Outbreak Identification

The identification of dust intrusions from the Sahara Desert (herein referred to as Saharan dust outbreaks (SDOs)) that affected the study area during 2015 was taken from the official information provided by the Spanish Ministry of Agriculture, Food and Environment each year [27], based on the methodology developed by this institution [28] and under European Commission guidelines [29]. The SDOs affecting the region were identified taking reference data from the European Monitoring and Evaluation Programme (EMEP) monitoring station in Barcarrota (southwest of Extremadura, LAT: 38°28′22″ N, LONG: 6°55′25″ W) and data from the rural background station in Monfragüe (northeast of Extremadura). Dates of the episodes produced during 2015 are shown in Appendix A (Appendix A). Figure 4 shows two EUMETSAT satellite images in which suspended dust from the Sahara Desert reaches the Iberian Peninsula from the south.

## 3. Results and Discussion

### 3.1. General Colour Features of PM Filters

Particulate matter (PM) captured on filters presents a wide range of colours depending on the concentration and nature of the collected material. The naked eye observation of PM_10_ filters obtained from the Badajoz and Monfragüe monitoring stations allowed us to identify different shades of grey, brown, orange and even black in some cases. Figure 5 shows the visualization of the colours of all filters collected at Badajoz (Figure 5a) and Monfragüe (Figure 5b) during 2015 and the corresponding gravimetric PM_10_ concentration values (*y*-axis), since each dot has its corresponding RGB values. For both locations, it was observed that at low PM_10_ concentrations the sample had a lighter colour that corresponded with high values of RGB, whereas an increase in particulate matter concentration caused the filter colour to become darker and the RGB values to decrease. This agrees with the fact that black colour is characterized by an R, G and B intensity of 0, 0, 0 values, whereas white colour presents values of 255, 255, 255 for R, G and B [30,31]. The green line in Figure 5 indicates the daily legal limit value of PM_10_ concentration for the protection of human health according to Directive 2008/50/EC (50 µg/m^3^), not to be exceeded more than 35 times a calendar year.

### 3.2. Overall Correlation Between Color Parameters and PM Concentration

The capability of the smartphone camera to estimate PM_10_ concentrations by digital image analysis of the material collected on the filters was first inspected by applying regression analysis between colour intensity values in RGB, HSV, HSL and greyscale models versus gravimetric PM_10_ concentration obtained by the standard methodology (described in Section 2.3). The analysis was performed for all samples collected at Badajoz and Monfragüe stations (total of 700 samples). Appendix A (Appendix A) show the mean values of colour intensity in the four colour models mentioned above, obtained from the image analysis of PM_10_ filters. The concentrations of particulate matter determined by the reference method for each filter and the sampling date are also indicated.

Table 1 and Table 2 show the regression parameters obtained for each digital image variable versus PM_10_ concentration. The *p*-values less than 0.05 for all of these parameters confirm the significant correlations between digital image parameters and PM_10_ concentrations. Negative slopes are due to the fact that the colour of higher load filters is closer to black (associated with RGB values closest to 0, 0, 0). In contrast, the lower load filters have a colour closest to white (associated with RGB values close to 255, 255, 255). In the case of saturation (S), which measures the relative purity degree or amount of white light of a colour, positive slopes are due to the fact that lower load filters have a colour closest to white or more discoloured (associated with S values close to 0%), whereas the colour of higher load filters is pure (associated with S values close to 100%). This fact shows that saturation increases linearly from white to pure colour [32].

Within all digital image variables, the saturation (S_HSV_) variable of the HSV model presented the highest coefficient of determination (R^2^ = 0.421), being the one that provides a better colour description of the sample filters. Good correlation between saturation and PM_10_ concentration is due to the fact that as the PM_10_ concentration increases, the density of the sample increases and the colour of the filters varies from very clear, or with a high amount of white light, to a colour of greater purity or low amount of white light, confirming that saturation increases linearly from white to pure colour, as we have commented previously. The other colour parameters, which mark the colour and intensity of it, showed lower correlation with PM_10_ concentration.

According to [33], the HSV model is an ideal tool for developing image processing algorithms based on natural colour descriptions for humans. On the contrary, the RGB model represents a colour as a mixture of three primary colours (red, green, blue), so it is ideal for generating an image colour but not for describing it. This is the reason that in the present colour analysis of PM_10_ samples on filters, better results were obtained with the S_HSV_ parameter than with the RGB model parameters, because a colour is not being generated but is describing a characteristic that changes with the sample (saturation, S). The RGB colour results from the superposition of three monochromatic images, so it is logical that R, G and B intensities can be extracted from an image, but their variation is independent of the sample concentration.

These results show that the proposed digital image analysis methodology applied to PM_10_ collected on filters is useful for estimating the concentration of particulate matter in ambient air based on the saturation (S_HSV_) colour parameter.

### 3.3. Influence of PM Source Variability on Colour Parameters

The proposed methodology of estimating PM_10_ concentration based on a digital image analysis of the collection filters was then tested for applicability to identify and estimate local and remote sources that affect the sampling locations at Badajoz and Monfragüe, and thereby determine the applicability of the proposed methodology to discriminate variable pollution events.

The correlation between the saturation (S_HSV_) variable obtained by the digital analysis of images versus the PM_10_ concentration measured by the standard method is shown in Figure 6. Interestingly, the samples that depart more from the linear correlation are those with the highest PM_10_ loads. These samples correspond to days when the region is under the influences of Saharan dust outbreaks (SDO days).

The main remote source affecting PM levels in Extremadura is the Sahara Desert, which periodically generates large masses of dust that are transported over long distances affecting Southwest Europe and even crossing the Atlantic Ocean. These Saharan dust outbreaks (SDOs) notably affect PM concentration and alter the usual colour of the samples captured on filters due to the characteristic mineral composition of this material. A new regression analysis was performed by disaggregating the data obtained for days under the influence of Saharan dust outbreaks (SDO days) and for days without influence from this remote source (No-SDO days). For the both types of days, the correlation between the saturation (S_HSV_) variable obtained by the digital analysis of images versus the PM_10_ concentration measured by the standard method is shown in Figure 7.

When regression analysis was performed, correlation parameters for the saturation (S_HSV_) of SDO days improved the results obtained previously for the whole data set to R^2^ = 0.599. Correlation parameters for No-SDO days were worse compared to those obtained for the entire data set (R^2^ = 0.300). These results indicate that the relationship between the S_HSV_ parameter and PM_10_ concentration departs from linearity probably due to the effect of the very different PM nature during SDO days. No-SDO days are characterized by a main source of great variability such as traffic, resulting in a very different colour pattern of the filters with respect to SDO days, where the predominant source was more homogeneous, and therefore, the colour of filters and the saturation parameter were more homogeneous throughout the data series. These results show that the application of the proposed digital image analysis methodology is more capable of estimating the PM_10_ concentration in ambient air on those days when the region was affected by dust intrusions from the Sahara Desert, according to the colour saturation that presents the particulate matter collected on filters.

On the other hand, as previously mentioned, Badajoz and Monfragüe present very different environmental features (mainly anthropogenic traffic sources in Badajoz, rural background sources in Monfragüe), which results in significantly different PM_10_ levels. For this reason, the influence of local sources in the applicability of the proposed digital analysis methodology to estimate the PM_10_ concentration was also studied, depending on the location of the sampling units, in addition to remote sources. A new regression analysis between the saturation parameter (S_HSV_) and the PM_10_ concentration was performed, separating the SDO days’ data in which they correspond to Badajoz and Monfragüe, and similarly it was performed with the No-SDO days’ data. Thus, four groups of data were obtained (SDO days of Badajoz, SDO days of Monfragüe, No-SDO days of Badajoz and No-SDO days of Monfragüe), each of which provided R^2^ parameters that described the correlation between the saturation of the sample and its concentration of PM_10_. In this way, both locations were compared for those days under the influence of Saharan dust outbreaks and for days when there was no remote source of pollution. For both locations and source, the correlation between the saturation (S_HSV_) variable obtained by the digital analysis of images versus the PM_10_ concentration measured by the standard method is shown in Figure 8 and Figure 9.

During SDO days, the correlation obtained between the S_HSV_ parameter and PM_10_ concentration is more significant in Badajoz than in Monfragüe, as it presents higher coefficients of determination (R^2^ = 0.646 for Badajoz versus R^2^ = 0.494 for Monfragüe) and a *p*-value less than 0.05. In the city of Badajoz, the Saharan dust outbreaks are more intense than in the National Park of Monfragüe, due to its location further south of the Iberian Peninsula, since these dust outbreaks reach the Iberian Peninsula from the south, as shown in Figure 4. This greater intensity means that PM_10_ concentrations of this nature are higher in Badajoz than in Monfragüe, giving the samples a purer and more homogeneous colour.

In the case of No-SDO days, the correlation obtained between the S_HSV_ parameter and PM_10_ concentration is again more significant in Badajoz than in Monfragüe, as it presents higher coefficients of determination (R^2^ = 0.372 for Badajoz versus R^2^ = 0.208 for Monfragüe) and a *p*-value less than 0.05. These results indicate that the digital image analysis provides a better performance when the PM_10_ concentration range is wider, as is the case of the urban location where the traffic source shows greater homogeneity. Monfragüe, being a natural park without local pollution sources, did not present samples with pure colour and, therefore, is not significantly related to colour saturation.

In view of the regression parameters obtained for each data set, which show the relationship between the saturation colour parameter (S_HSV_) and PM_10_ concentration, the most significant correlation is the one related to the PM_10_ samples obtained in Badajoz station for SDO days, as evidenced by the highest coefficient of determination, namely R^2^ = 0.646. This result indicates that the proposed digital image analysis methodology can be in principle more appropriate for estimating PM_10_ levels under high PM_10_ variability due to remote sources in locations with higher pollution levels such as urban environments.

### 3.4. Source Assignment Based on Colour Parameter Analysis

#### 3.4.1. Local Sources at Sampling Point

As previously mentioned in Section 3.3, Badajoz and Monfragüe present very different environmental features, which translate into significantly different levels of PM_10_. Therefore, we tested the capability of the proposed digital analysis of the S_HSV_ parameter to characterize PM_10_ differences between both sites during 2015. Figure 10 presents the comparison of S_HSV_ values for the images of the filters sampled in Badajoz and Monfragüe. Both locations showed asymmetric data distributions since the median is displaced towards the lower limit of the box, where values tend to concentrate (Shapiro-Wilk test, *p* < 0.05). In Monfragüe, 75% of values lie between 3.5% and 13%, whereas the same percentage in Badajoz is located between 2% and 20%. The results agree with the fact that PM_10_ levels in Monfragüe are lower than in Badajoz (as confirmed by gravimetric results), so the colour saturation of particulate matter in Monfragüe is closer to white (lower percentage of pure colour), indicating lower pollution levels, whereas the values of Badajoz are closer to pure colour. The distribution of the S_HSV_ parameter obtained in Badajoz had greater dispersion than those obtained in Monfragüe. This is probably due to the wider range of anthropogenic source variability in Badajoz compared with the more stable natural sources affecting PM_10_ levels in Monfragüe National Park.

Statistical differences in S_HSV_ values between the two locations studied were tested by the Mann–Whitney Test. The *p*-values obtained were less than 0.0001, which confirms that S_HSV_ values obtained in Badajoz are significantly different from those obtained at Monfragüe.

#### 3.4.2. Remote Sources

As previously mentioned, the main remote pollution source in the Extremadura region is the Saharan dust outbreaks. This source alters the usual colour of the samples captured on filters due to the characteristic mineral composition of this material. We explored the capabilities of the digital image analysis of the filters to identify the presence of SDO at each sampling location (Badajoz and Monfragüe). Figure 11 shows the comparison of S_HSV_ values for samples taken during No-SDO days versus the values for samples taken during SDO days at Badajoz and Monfragüe. In the Badajoz sampling location, the S_HSV_ distributions showed similar dispersion for No-SDO days and for SDO days because the samples of both days presented a similar homogeneity. They only differentiated in that, for the SDO days, 75% of the S_HSV_ data are between 4% and 39%, whereas for the No-SDO days, 75% of the data are between 2% and 33%. This is because Saharan dust outbreaks probably mask local pollution sources, and a sample with greater colour purity was obtained (higher saturation percentage value). In the case of Monfragüe, the S_HSV_ distributions showed a greater dispersion for SDO days than for No-SDO days. This was due to the fact that, on SDO days, the samples presented a wide range of PM_10_ concentrations depending on the intensity of the dust each day, which entails different degrees of colour sample purity. On the other hand, on No-SDO days, the samples were affected by the low variability and same source intensity, causing the samples to always have a very similar colour purity, and saturation values did not present a great variability.

The Mann-Whitney Test confirmed the existence of significant differences in S_HSV_ values for SDO days versus No-SDO days at both locations. In Badajoz, the S_HSV_ variable gave a *p*-value of 0.0001, which is less than 0.05 indicating significant differences between SDO and No-SDO days. In the case of Monfragüe, the S_HSV_ variable provided a *p*-value of less than 0.0001, so the values are more sensitive for identifying differences between SDO and No-SDO days than in the case of Badajoz. The lowest *p*-value confirmed that the S_HSV_ digital image variable is remarkably appropriate for identifying PM_10_ level variations.

Based on the Figure 11 box-plot, both locations were also compared on SDO days and No-SDO days. In the case of SDO days, Badajoz had a slightly higher dispersion than Monfragüe, although the range of S_HSV_ values in which 75% of the data was found was similar. In contrast, on No-SDO days, the dispersion was much greater in Badajoz than in Monfragüe, as is the range of S_HSV_ values. The explanation for these results was based on the fact that the SDO days samples have the same nature, so that the differences that may exist are due to source intensity. In contrast, for No-SDO days, the pollution sources have different origins and the samples do not have similar characteristics as for the SDO days. The Mann-Whitney test application for the S_HSV_ values of Badajoz and Monfragüe for SDO days resulted in a *p*-value of 0.145, confirming that there are no significant differences between both data sets when the locations are affected by Saharan dust outbreaks. Meanwhile, for No-SDO days, the *p*-value obtained was less than 0.0001, indicating that when pollution is due only to local sources, there are significant differences between both locations.

The results showed that the application of the proposed digital image analysis methodology is more capable of identifying the influence of the SDO remote pollution source in the rural environment of Monfragüe than in the urban environment of Badajoz. This behaviour can be related to the fact that SDO episodes are somewhat masked in Badajoz due to the combination with local sources of anthropogenic pollution, so the smartphone camera is less capable of detecting the reduced differences in colour. On the other hand, colour differences between SDO and No-SDO days are more easily detected by the smartphone camera in the Monfragüe samples, which are only affected by natural PM sources without the colour interference from local anthropogenic sources.

## 4. Conclusions

In this study, the potential applicability of a novel methodology to monitor the levels of particulate matter in ambient air, based on the digital analysis of images of PM_10_ samples captured on glass fibre filters after high volume capture in field units, was demonstrated on real samples from the surveillance network of the air quality of Extremadura (Spain). The images were obtained using the camera of a conventional smartphone in a box with reproducible LED lighting and a software application for image analysis. The obtained results confirm that the proposed methodology allows users to correctly estimate the concentration of particulate matter in the ambient air, through a significant correlation between the RGB/HSV/HSL/greyscale colorimetric parameters and the experimental concentration of PM_10_ (gravimetric analysis), the saturation parameter (S_HSV_) being the one that provides a better colour discrimination ability of the sample filters. The ability of the proposed methodology to differentiate groups of samples from environments with different environmental characteristics in terms of local sources of particle emissions to the ambient air (urban location versus rural location) was also demonstrated. The smartphone camera showed a better discriminatory capacity regarding local sources in the case of the urban environment, probably because of the wider range of PM_10_ concentrations observed in such an environment. The methodology is also able to identify the influence of remote sources of particles such as dust intrusions from the Sahara Desert, which episodically affect the study area. In this case, the proposed methodology was more reliable to identify Saharan air intrusions in the samples coming from the rural environment of the Monfragüe National Park than in the case of the city of Badajoz, probably due to the greater interference of anthropogenic local sources on the colour of the samples in the urban environment. The methodology of digital analysis of filter pictures can be considered a promising first approach towards a system for measuring the concentration of PM in ambient air, a system that is simpler, faster and more miniaturized than conventional ones. The system could be coupled to a portable sampling system and then added to any digital image capture device, so that anyone can obtain photographs of the captured PM and thus estimate a concentration range in real time and take protective measures if necessary, complementing the information provided by the official environmental monitoring networks.

## Figures and Tables

**Figure 1 sensors-19-04791-f001:**
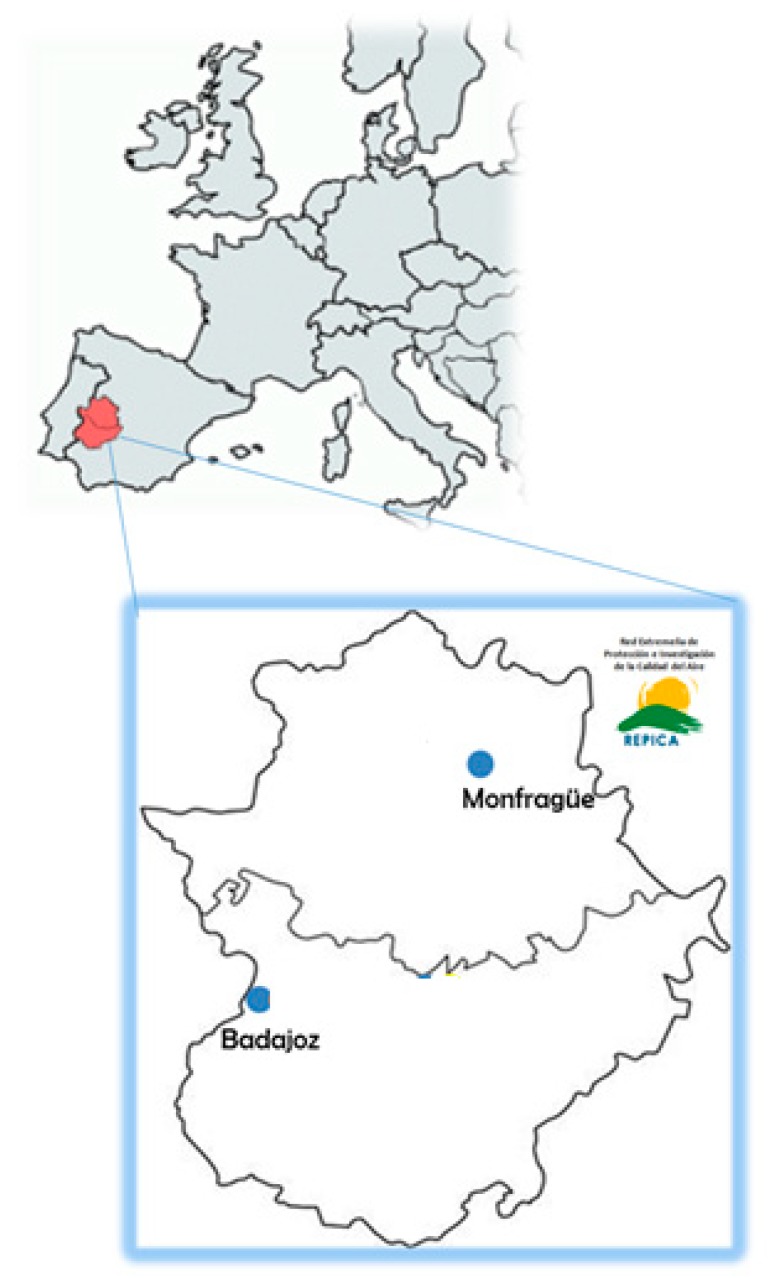
Map of the PM_10_ sampling monitoring stations in the Extremadura region (Spain).

**Figure 2 sensors-19-04791-f002:**
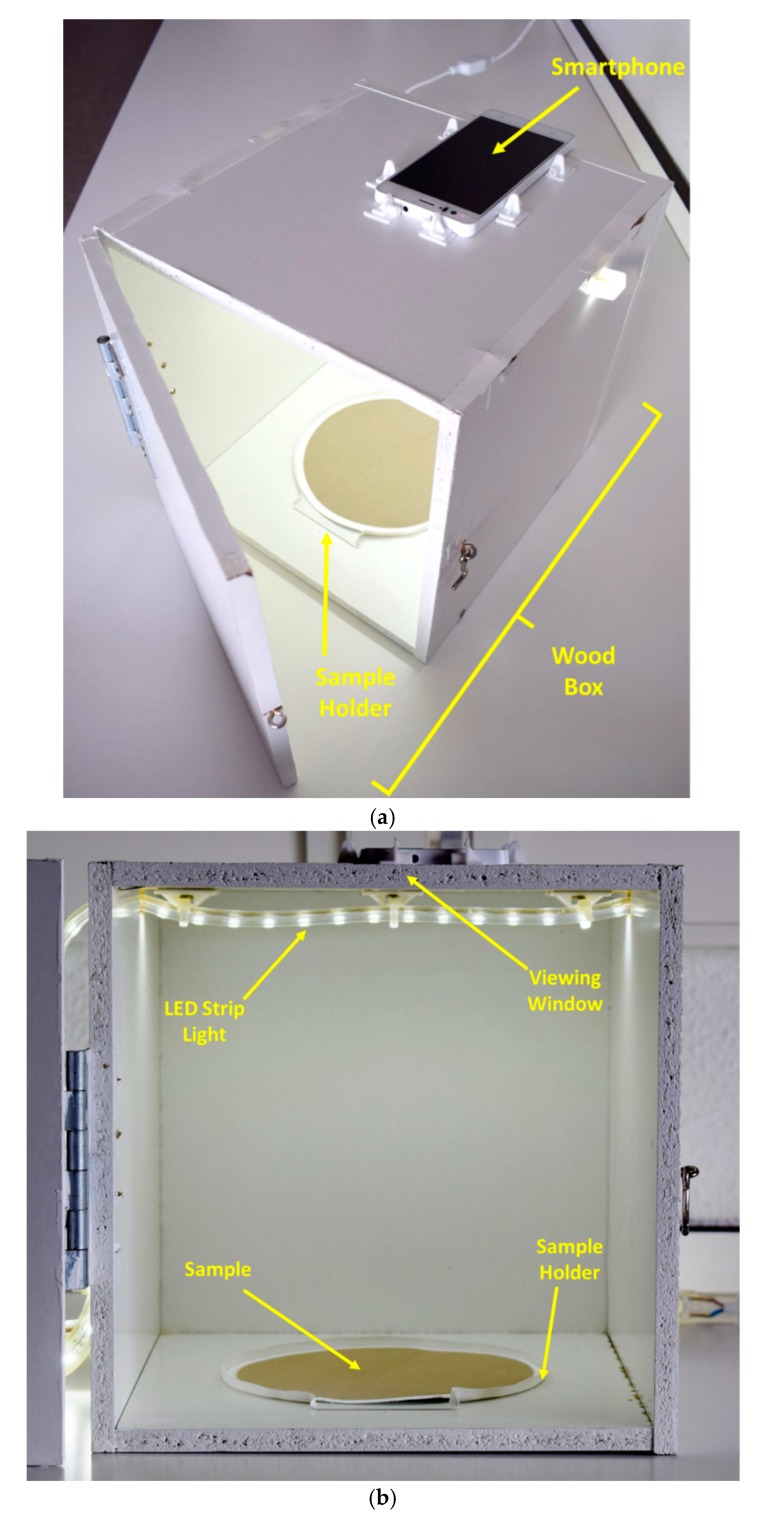
(**a**) Home-made low-cost photographic system for monitoring PM samples; (**b**) view of the sample holder and LED strip light source; (**c**) a PM sample image obtain with the system.

**Figure 3 sensors-19-04791-f003:**
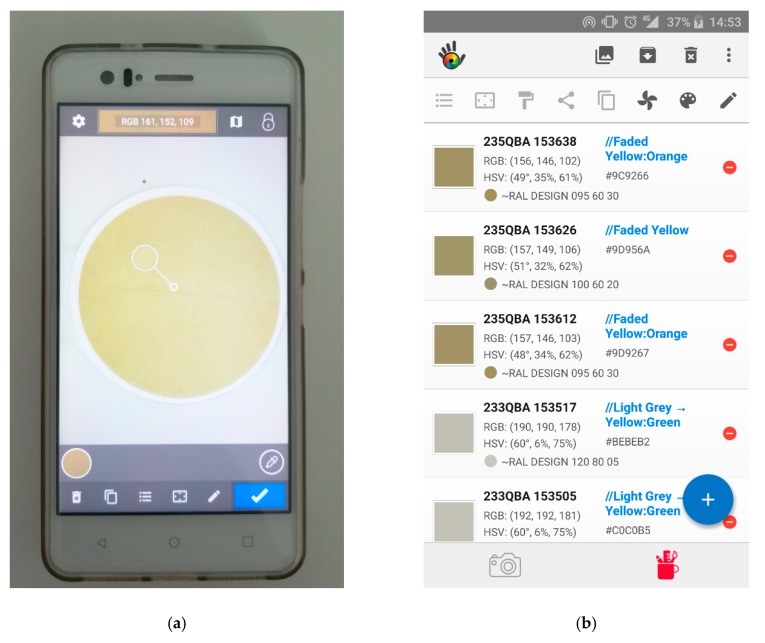
(**a**) Central region of image selected by the Color Grab app; (**b**) red, green, blue (RGB) and hue, saturation, value (HSV) information extracted from the Color Grab app.

**Figure 4 sensors-19-04791-f004:**
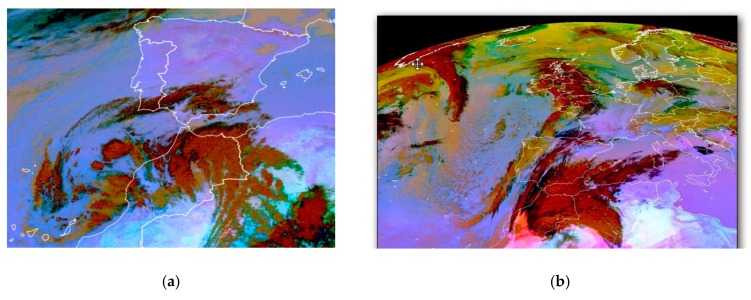
RGB “dust” composition. Second Generation Meteosat, EUMETSAT. The area embossed in red shows the suspended dust located on the African continent and the south of the Iberian Peninsula, associated with a storm. (**a**) Saharan dust outbreaks on January 31, 2018; (**b**) Saharan dust outbreaks on 17 February 2014.

**Figure 5 sensors-19-04791-f005:**
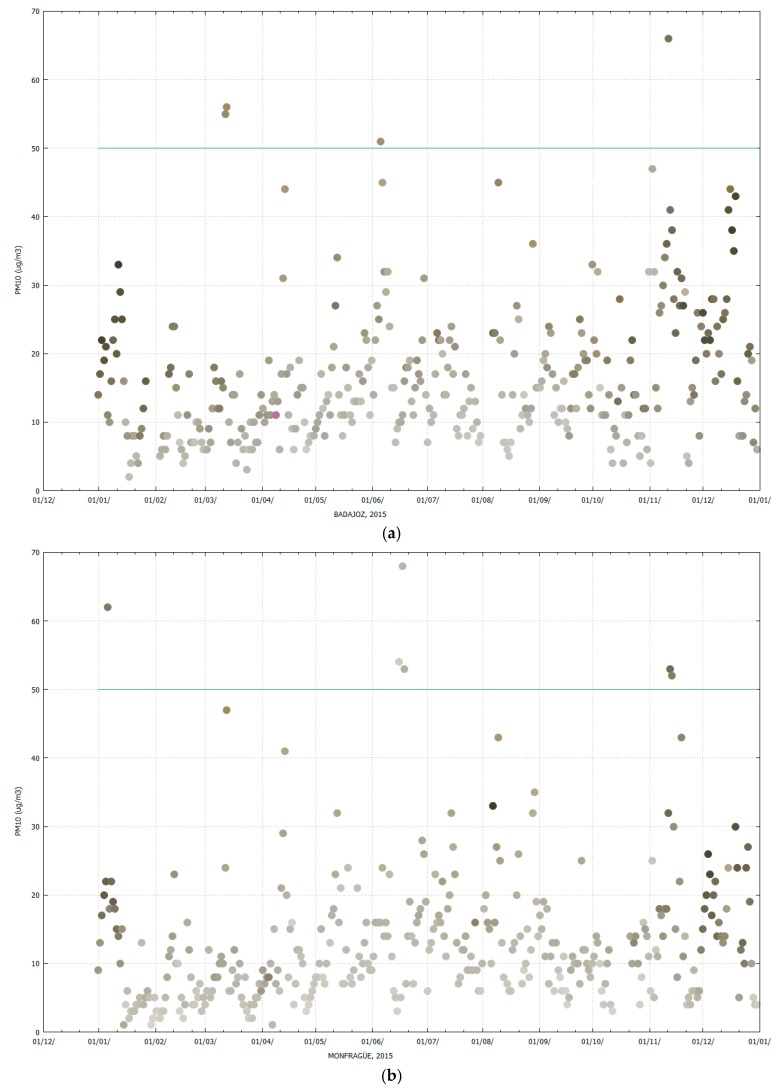
Evolution of PM_10_ concentration during 2015 (**a**) Badajoz (BA); (**b**) Monfragüe (MF). The colour of each sample dot corresponds to its RGB value. The green line represents the legal daily limit value of PM_10_ concentration for the protection of human health according to Directive 2008/50/EC (50 µg/m^3^).

**Figure 6 sensors-19-04791-f006:**
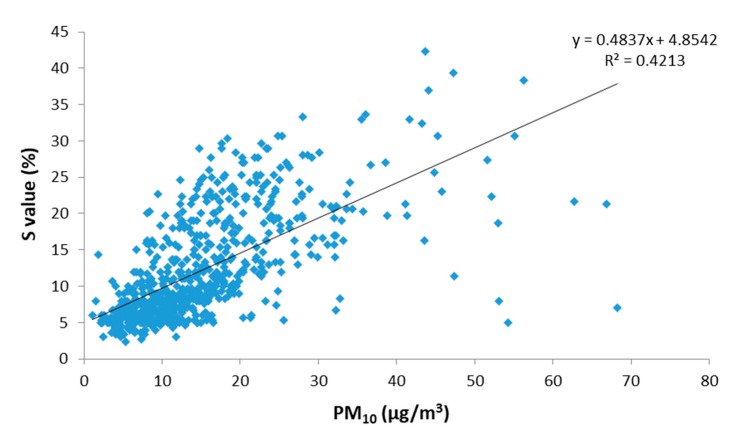
Relationship between saturation (S_HSV_) and gravimetric PM_10_ concentration for all daily samples collected during 2015 at Badajoz and Monfragüe (total of 700 samples).

**Figure 7 sensors-19-04791-f007:**
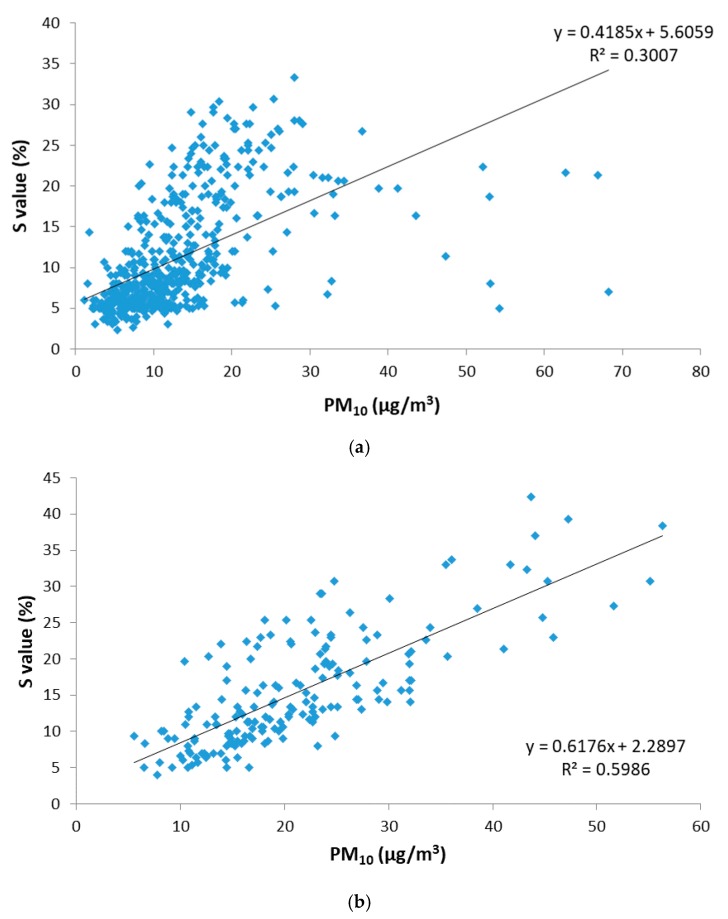
Relationship between saturation (S_HSV_) and gravimetric PM_10_ concentration for remote source influence during 2015 at Badajoz and Monfragüe. (**a**) No-SDO days and (**b**) SDO days.

**Figure 8 sensors-19-04791-f008:**
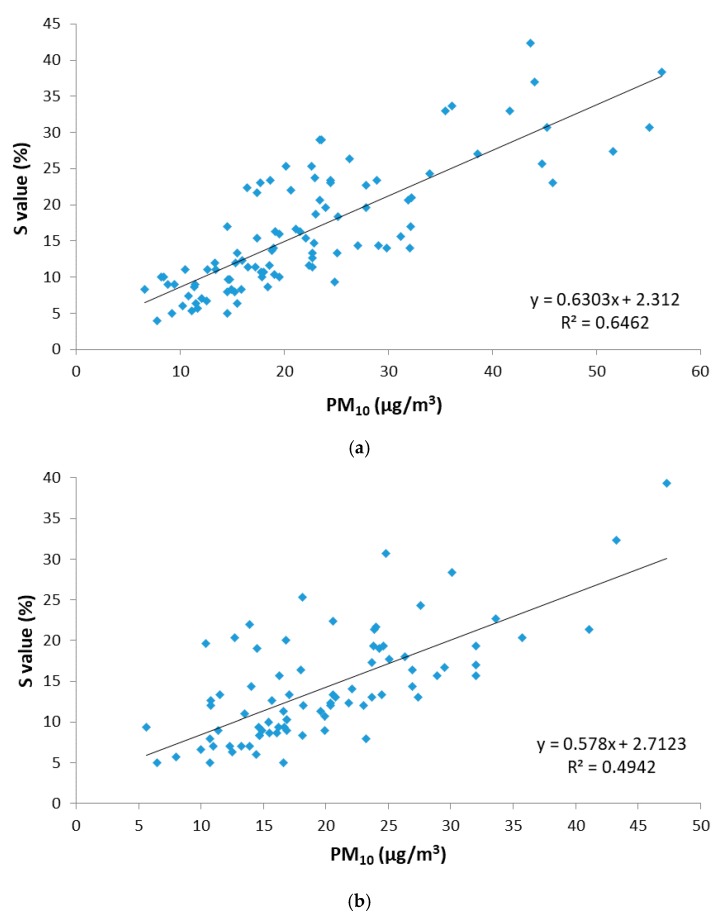
Relationship between saturation (S_HSV_) and gravimetric PM_10_ concentration for days under remote source influence (SDO days) during 2015. (**a**) At Badajoz location; (**b**) at Monfragüe location.

**Figure 9 sensors-19-04791-f009:**
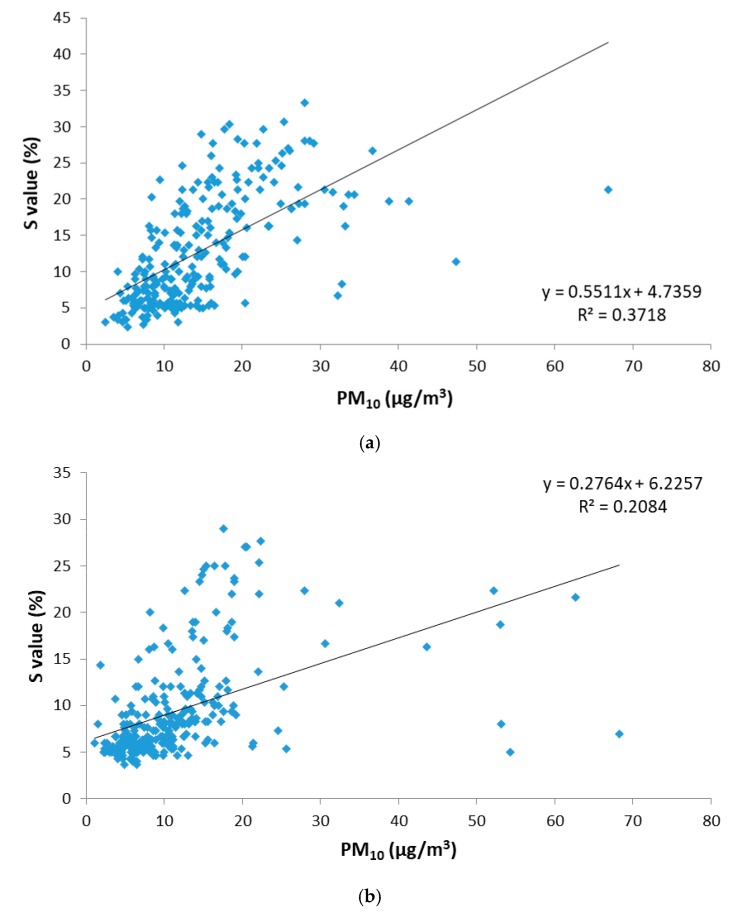
Relationship between saturation (S_HSV_) and gravimetric PM_10_ concentration for days under local source influence (No-SDO days) during 2015. (**a**) At Badajoz location; (**b**) at Monfragüe location.

**Figure 10 sensors-19-04791-f010:**
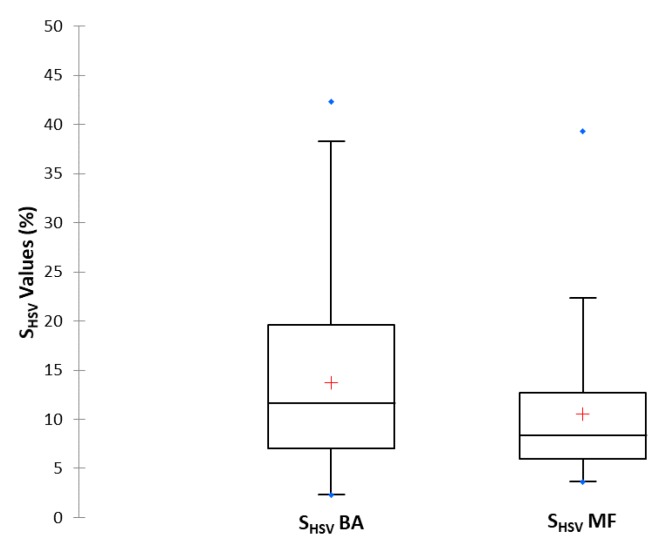
Box-plot of S_HSV_ values obtained from Badajoz and Monfragüe monitoring stations for total days of 2015.

**Figure 11 sensors-19-04791-f011:**
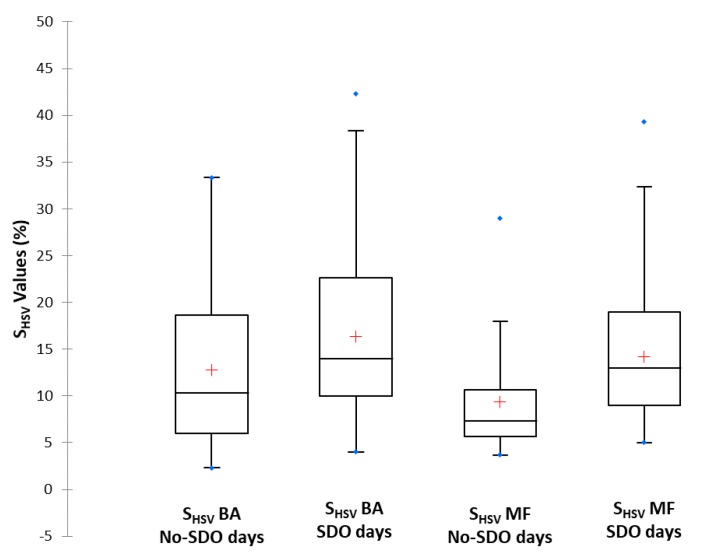
Box-plot of S_HSV_ values obtained to compare values for No-SDO days versus SDO days from Badajoz and Monfragüe monitoring stations.

**Table 1 sensors-19-04791-t001:** Linear regression parameter between R, G, B, H_HSV_, S_HSV_, V and PM_10_ concentrations for all samples collected in Badajoz and Monfragüe during 2015.

	Red (R)	Green (G)	Blue (B)	Hue (H_HSV_)	Saturation (S_HSV_)	Value (V)
Coefficient of Determination, R^2^	0.286	0.302	0.350	0.042	0.421	0.287
Intercept	192.594	191.143	181.990	51.703	4.854	75.560
Slope	−1.639	−1.757	−2.161	−0.085	0.483	−0.646

**Table 2 sensors-19-04791-t002:** Linear regression parameter between H_HSL_, S_HSL_, L, Lu, Li, Avg and PM_10_ concentrations for all samples collected in Badajoz and Monfragüe during 2015.

	Hue (H_HSL_)	Saturation (S_HSL_)	Luminance (L)	Luminosity (Lu)	Lightness (Li)	Average (Avg)
Coefficient of Determination, R^2^	0.028	0.333	0.321	0.303	0.322	0.316
Intercept	51.880	7.686	73.465	190.807	187.252	188.576
Slope	−0.090	0.180	−0.746	−1.761	−1.899	−1.853

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
