# Peer review of "Estimation of PM10 Levels and Sources in Air Quality Networks by Digital Analysis of Smartphone Camera Images Taken from Samples Deposited on Filters"

_sensors, 2019, doi:10.3390/s19214791_

Round 1
Reviewer 1 Report
The authors have addressed the points raised by the reviewers and the paper has improved substantially. I recommend the paper is accepted.
There are a few minor spelling errors remaining. Further, the reported p-values do not seem to make much sense (such low values are typically not reported). The authors are advised to consult a statistician on this.
Reviewer 2 Report
Thank you for submitting your manuscript. The overview of current technologies is good. Overall organization is good but could use some general editing. The paper gets somewhat harder to read after the first 10 pages or so.
It might be helpful to add references in lines 44 and 45. Line 57: “to decentralized” consider the grammar of the sentence. Line 147: “strip light that simulates daylight”. What is the significance of daylight simulation? How does different types of particles affect the color of the filter? With p values < 1E-10, what is its significance? How much of it is due to collection or processing error? Lines 270-272: “Rest of colour parameters, which mark the colour and intensity of it, did not show good correlations because in the samples these characteristics did not change, confirming that their variation is independent on the PM10 concentration.” What is the reason that intensity of a color components didn’t change with PM10 concentration? Generally, when p-value is used, the null hypothesis was not clearly stated, making it a little hard to evaluate. How does seasonal changes affect the data? Were the tests carried out in dry or wet weather? Inconsistency using “No-DSO”, “NO-SDO”, and “Non-SDO”.Author Response
Please see the attachment

This manuscript is a resubmission of an earlier submission. The following is a list of the peer review reports and author responses from that submission.
Round 1
Reviewer 1 Report
This paper explores the performance of smartphone cameras as low cost and easily accessible tools to provide information about the levels and origin of particulate matter in ambient air. PM10 levels have been evaluated by studying images of daily PM10 samples captured on glass fibre filters. This work is interesting for readers in the field of atmospheric environmental monitoring. However, there are a few concerns, which should be addressed before the paper published.
(1) The authors claim one novelty of the present manuscript is to utilize a smartphone. It is however not advantageous from my point of view. One can acquire images through a very simple CCD camera and then perform post analysis in an embedded system (e.g., Arm chip). The authors should discuss more about the advantages of utilizing a smartphone.
(2) The Scheimpflug lidar technique has also utilized a camera to acquire backscattering image from a transmitted cw laser beam, from which the atmospheric extinction coefficient can also be evaluated. The authors should reference and discuss a bit in their introduction (e.g., Optics Express 25, A628-A638,2017; Optics Express 26, 31942-31956,2018).
(3) Is there an APP for image processing to show the PM10 level? If yes, a picture to show how the APP looks like could be nice.
(4) It would be nice to discuss the possibility of adding a polarization detection channel in the image analysis method, e.g., use a second camera to take a depolarized image.
(5) It seems that the correlation between the results obtained by the smartphone and the PM10 concentration was not very high. How could this be useful for PM10 level monitoring? The result could also be dependent on the relative humidity. The authors should be able to eliminate its influence by taking data from metrological stations.
(6) In the end of the manuscript, the authors claim that “The system could be coupled to a portable sampling system and then added to any digital image capture device, so that anyone can obtain photographs of the captured PM and thus estimate a concentration range in real time and take protective measures if necessary, complementing the information provided by the official environmental monitoring networks.” This is too exaggerated. Although smartphone might be available for everybody, the technique also require a well-controlled sampling system, which would not be easy for “anyone” to accomplish.
Reviewer 2 Report
The manuscript "Estimation of PM10 levels and sources in air quality networks by digital analysis of smartphone camera images taken from samples deposited on filters" by Carretero-Peña et al presents a valuable proposal for the estimation of particle matter levels. The method is well described, the study has been done carefully and the interpretation makes sense. The method is also potentially interesting for other researchers.
The paper is adequate for the journal scope, it is well-formatted and presents a valuable experiment, so it deserves publication in this Journal.
I can only congratulate the authors for the excellent work.
However, before accepting the manuscript for publication, I have a few suggestions to correct some small details.
Minor changes:
- pag. 4, line 146: "... in order to provide stable...";
- pag. 4, line 148: please use lumen instead of lumens;
- pag. 4, line 148: 6500 K
- all over the manuscript: please consider using nonbreakable spaces between numbers and their units;
- pag. 7, line 189: "... and superior for the G...";
- pag. 9, line 236: "Tables S1-S2 show the...".
Reviewer 3 Report
The authors present a method to use a smartphone in combination with a filter and a LED light source to estimate PM10 levels in urban and rural areas in Spain. The presented method is simple and cheap but the R2 values for all tested combinations (R/G/B/Lu/Li/Av) are low limiting the usefulness of the method. Therefore it can expected that suggestions for further improvements are given, yet in the results and discussion section there is little/no discussion on how the system can be further improved in order to obtain more reliable measurements. Further, the data analysis is somewhat one sided focusing heavily on the correlations with the parameters (R/G/B/Lu/Li/Av).
Other remarks:
Is there any correction made for changes in light intensity or changes in spectral distribution of the LED light source?
Does a combination of parameters provides a better correlation than a single parameter?
Check the number of significant digits in Tables 1-3.
Figure 4: used font on the x-axis and y-axis too small
First time box plots are presented (Figure 6) provide some explanation.
Figure 7: outlier box plot Blue (B) hidden in the text
In the introduction the authors could include Nyarku, M., et al. (2018). Mobile phones as monitors of personal exposure to air pollution: Is this the future?. PloS one, 13(2), e0193150.
Some spell check is needed (e.g. lines 164, 209)
Round 2
Reviewer 1 Report
The correlation between the results obtained by the smartphone and the
PM10 concentration was still very low, and the authors do not provide sufficient information to state that the performance can be further improved. One can simply buy a PM10/PM2.5 instrument (50 dollars) based on the light scattering method, which will have much better performance than the present approach. There are also numerous literatures about this method. Thus, I could not see any novelty of the present work.
Reviewer 3 Report
The quality of the PM measurements with the camera is very low which currently strongly limits the impact of the developed method. Albeit such a measurement system is relatively cheap, no proper suggestions further improvement of the system are given in the revised version and therefore I am ambivalent about the publication of the paper.